# Attention-Based Joint Training of Noise Suppression and Sound Event Detection for Noise-Robust Classification

**DOI:** 10.3390/s21206718

**Published:** 2021-10-09

**Authors:** Jin-Young Son, Joon-Hyuk Chang

**Affiliations:** Department of Electronic Engineering, Hanyang University, Seoul 04763, Korea; ekwy3tp@hanyang.ac.kr

**Keywords:** noise-robust classification, noise suppression, sound event detection, joint training, deep neural network, attention

## Abstract

Sound event detection (SED) recognizes the corresponding sound event of an incoming signal and estimates its temporal boundary. Although SED has been recently developed and used in various fields, achieving noise-robust SED in a real environment is typically challenging owing to the performance degradation due to ambient noise. In this paper, we propose combining a pretrained time-domain speech-separation-based noise suppression network (NS) and a pretrained classification network to improve the SED performance in real noisy environments. We use group communication with a context codec method (GC3)-equipped temporal convolutional network (TCN) for the noise suppression model and a convolutional recurrent neural network for the SED model. The former significantly reduce the model complexity while maintaining the same TCN module and performance as a fully convolutional time-domain audio separation network (Conv-TasNet). We also do not update the weights of some layers (i.e., freeze) in the joint fine-tuning process and add an attention module in the SED model to further improve the performance and prevent overfitting. We evaluate our proposed method using both simulation and real recorded datasets. The experimental results show that our method improves the classification performance in a noisy environment under various signal-to-noise-ratio conditions.

## 1. Introduction

Sound event detection (SED) aims to detect the type of event corresponding to an incoming sound and to obtain its onset and offset. SED is applied to various fields, and with the development of technology, it is commonly used in fields closely related to human lives. For instance, it is being used in automatic assistance driving [1], smart meeting rooms [2], drone detection [3], multimedia [4], social care [5], audio surveillance system [6].

Early research in the field of SED used traditional shallow learning model approaches, such as Gaussian mixture models [7], hidden Markov models [8], and random regression forests [9]. Approaches based on support vector machines [10,11,12] and non-negative matrix factorization [13,14,15] were also proposed. In recent years, deep neural network (DNN)-based approaches such as convolutional neural network (CNN) [16,17,18], recurrent neural network (RNN) [19,20], and convolutional recurrent neural network (CRNN) [18,21] have presented high classification performance.

The above-mentioned recent studies were focused on improving the SED performance, which demonstrated its potential for applications in real environments. However, such applications are affected by ambient noise and cannot detect and classify the desired target sound. Therefore, it is important to overcome the interference of a noise signal. In this regard, in some fields, studies focused on the characteristics of noise have been conducted. Some studies focused on the point that noise components have non-Gaussian properties in communication and radar systems and suggested solutions related to this, and a study to improve the robustness of the DOA estimation against the interference and noise signals was conducted [22,23,24]. Research related to noise-robust SED also has been performed as the growing importance of being robust in noise signals for application in a real environment. Methods such as weak-level noise reduction approaches [25], adaptive noise reduction [26], and optimally modified log-spectral amplitude-based noise filtering [27] were presented. In addition, studies combining DNN-based dereverberation and beamforming at the front end [28] were conducted. However, there is limited research on noise-robust SED using DNN-based audio enhancement at the front end. In particular, a good-performance time-domain audio enhancement model, such as convolutional time-domain audio separation network (Conv-TasNet) [29], has not been studied for use at the front end of SED. However, in the field of automatic speech recognition (ASR), various studies have been conducted on combining DNN-based speech enhancement at the front end with DNN-based ASR at the back end in the time–frequency domain to develop a noise-robust ASR system [30,31,32,33,34]. In the time-domain, for enhancing the noise-robust performance by speech denoising, combining Conv-TasNet at the front end was proposed [35]. Furthermore, using a joint fine-tuning method after combining with Conv-TasNet was suggested to improve the performance of music-mixed speech recognition [36].

Motivated by the joint DNN-based audio enhancement actively conducted in the ASR field, this paper proposes for the first time in the field of SED combining DNN-based time-domain noise suppression (NS) at the front end to increase the SED performance in a low signal-to-noise ratio (SNR) environment. For the NS model at the front end, we use a temporal convolutional network with the group communication with context codec method (GC3-TCN) [37], which reduce the model complexity of Conv-TasNet and secures the same performance. In [37], time-domain GC3-TCN was originally used for audio and speech separation; however, in this study, it is modified for NS. For the SED model at the back end, a CRNN-based classification model is employed. Subsequently, the pretrained NS and SED models are cascaded, which are trained for different objectives, and a joint fine-tuning method by learning with the final SED loss is proposed. During the joint fine-tuning process, we propose to freeze some layers’ weight and add an attention module to prevent overfitting and improve the classification performance in a noisy environment. The proposed method is evaluated using simulation and real recorded data.

The remainder of this paper is organized as follows: Section 2 describes the proposed system, which is composed of building pretrained NS and SED models, a joint training procedure, and an attention mechanism. The experimental settings and the evaluations are presented in Section 3. Section 4 discusses the experimental results. Finally, in Section 5, the conclusions are drawn.

## 2. Proposed System

In this section, we describe our proposed system, which consists of pretraining modules of each NS and SED, a joint training module of combined two models, and an attention module used in the joint training process. Prior to the fine-tuning by joint training, the pretrained models are NS and SED models, as shown in Figure 1. Each pretrained model is used to create a deeper DNN model, an attention module is added in the joint training stage, and fine-tuning is performed based on the SED loss. An input signal *X* undergoes the following process:(1)$$X_{enh}=NS\left(X\right)$$
(2)$$\hat{X}=FE\left(X_{enh}\right)$$
(3)$$Y^{classifier}=SED\left(\hat{X}\right)$$
where $X_{enh}$ denotes the enhanced waveform, FE(·) is the feature extraction, $\tilde{X}$ denotes extracted log-mel spectrogram feature and $Y^{classifier}$ is the classifier output.

### 2.1. Noise Suppression Model

For the NS model at the front end, which is combined with the SED model at the back end, the GC3-TCN method is applied. Its key concept is sub-band processing, which divides the intermediate representations into a specific number of feature groups and processes them separately. It reduces the model size and complexity by weight sharing across all groups (group communication), and further decrease the number of multiply-accumulate operation using encoder-decoder-based temporal compression method (context codec). In the encoder part of the context codec, the temporal context of local feature is summarized into a single feature representing the global characteristics of the context [37]. After passing the group communication-equipped separation module, the compressed feature is transformed back to the context feature through the decoder part of the context codec and reconstructed to the estimated waveforms through a decoding transformation. Considering the model size of the joint model consisted of NS and SED, we used the GC3-TCN for the NS model. More details are described in [37].

#### 2.1.1. Deep Feature Loss

The NS model is trained using deep feature loss. In [38], a pretrained auxiliary model was used to train the enhancement model. Similar to the method in [38], in the NS model training process in this study, a pretrained auxiliary network that was trained with a clean dataset for classification is introduced and its weights are frozen. The deep feature loss is the $L^{1}$ loss in the difference between the activations of the clean input feature and the predicted enhanced feature (that undergoes NS at the front-end process) through each layer in the auxiliary model as follows:(4)$$L\left(s,n\right)=\sum_{m=1}^{M}{∥\Omega^{m}\left(F\left(s\right)\right)-\Omega^{m}\left(F\left(N\left(n\right)\right)\right)∥}_{1}$$
where *s* and *n* denote the clean and noisy input signals, respectively. In addition, *N*(·) is the operation in the NS model, *F*(·) is the feature extraction, *M* is the number of layers, and $\Omega^{m}$(·) is the activation feature output of the $m^{th}$ layer in the auxiliary model.

#### 2.1.2. Auxiliary Model

The auxiliary model used for training the NS model is a CRNN-based classification model, which is the same as the SED model at the back end with only differences in the clean and noisy input features as shown in Figure 1. The model is trained with a clean log-mel spectrogram feature, and its weights are frozen in the NS model training procedure. Three convolutional layers with learnable kernel sizes of 3 × 3 are used to learn high-level feature representations from the log-mel spectrogram feature. Each layer is followed by a max-pooling layer with a 3 × 3 window size. The feature passes through each convolutional layer followed by a max-pooling layer and is subsequently fed into three bidirectional long short-term memory (Bi-LSTM) layers, which are used to capture the temporal context dependencies. Finally, the feature is fed into a fully connected layer and a sigmoid activation layer to obtain the event presence probability output.

### 2.2. Sound Event Detection Model

A CRNN-based classification model is used for the SED model, same as the auxiliary network used in the NS model training procedure, as described above. A noisy log-mel spectrogram feature is fed into the CRNN model, and the output is the event presence probability for the same number of sound event classes as that in the dataset. The label is one-hot encoded; therefore, the output range is [0, 1].

### 2.3. Joint Training

We propose joint fine-tuning to update all parameters except the last fully connected layer by combining the GC3-TCN-based NS model and the CRNN-based SED model. The overall structure is a network that simply cascades the pretrained NS and SED models, as shown in Figure 2. In the input/output process, the input mixture waveform enters the NS model to yield an estimated noise suppressed waveform output (blue section in Figure 2), and after conversion into a log-mel spectrogram by feature extraction (purple section in Figure 2), it is fed into the SED model to yield the event presence probability output (orange section in Figure 2). In the joint training procedure, the loss is propagated down from the back end to the front end by setting the SED loss as the loss of the combined network.

### 2.4. Attention Mechanism

In the joint training process, an attention mechanism is added between the input in the SED module and the output passing through each convolutional layer and max-pooling layer. We exploit an attention mechanism of a similar form used in [39,40], but with different stride sizes of each convolutional layer, normalization, and skip-connection. As shown in Figure 3, *i* is set as a variable representing the order of blocks (convolutional and max-pooling layers) to which the attention module is added. The input feature and the $i^{th}$ feature from the $i^{th}$ convolutional and max-pooling layer output are mapped to the two-dimensional (2-D) feature through a 2-D convolutional layer with a 3×3 kernel size. The 2-D features each obtained from different inputs, the input and $i^{th}$ output in Figure 3, have the same dimensions as the $i^{th}$ output, by setting different stride sizes for the conv2D layer before they are added. An attention mask is produced after passing through the sigmoid activation, 2-D convolutional layer, and another sigmoid activation. The feature is subsequently element-wise multiplied with the obtained attention mask, and the masked feature is added to the $i^{th}$ convolutional and max-pooling layer output.

## 3. Experiment

### 3.1. Dataset

The proposed system was evaluated using a mixture generated by mixing clean sound events data with the noise data. In addition, to verify the SED performance in a real environment, not only simulated noise data but also noise data recorded in a real environment were used. As for sound events audio data, we used the TAU Spatial Sound Events 2019 dataset [41]. The dataset consists of ambisonic and microphone array datasets. The ambisonic dataset contains four-channel first-order ambisonic recordings, and the microphone array dataset comprises four-channel directional microphone recordings from a tetrahedral array configuration [41]. In this study, we used the microphone array dataset. The microphone array dataset consists of a total of 500 audio data, 400 for development, and 100 for evaluation. Each audio clip is 1-min long and has a sampling rate of 48 kHz. The recordings were synthesized using spatial room impulse response collected from five indoor locations, with 504 unique azimuth–elevation–distance combinations [41]. The IRs were convolved with isolated sound events from DCASE 2016 task 2 dataset. The dataset consists of 11 classes such as clearing throat, coughing, door knock, door slam, drawer, human laughter, keyboard, keys (put on table), page turning, phone ringing, and speech. In this study, the development dataset was used, and each 60-s audio clip was divided into 10-s audio clips. Prior to the joint training, the divided dataset was used as the NS model, auxiliary network, and SED model dataset. In the joint training stage, the dataset was divided into 5-s audio clips to double the total dataset.

For the simulated noise data, we used the DNS Challenge Noise-full band dataset [42], which were selected from Audioset [43] and Freesound. The dataset contains approximately 150 audio classes, such as music, speech, toothbrush, creak, etc., and 60,000 clips from Audioset and additional 10,000 noise clips from Freesound and the DEMAND dataset [44]. Each audio clip is 10-s long and has a sampling frequency of 48 kHz.

For the real experimental noise data, we recorded a real sound source in a noisy environment created using a robot vacuum cleaner manufactured by LG Electronics. We mounted a four-channel microphone array on a robot in the form of a tetrahedron, and only the single channel closest to the robot was used for the experiment. In the recording environment, the robot cleaner moved freely, and there were two modes and an additional turbo on/off mode; therefore, a total of four types of vacuum noises were recorded. All signals were recorded at a sampling frequency of 48 kHz. To construct a dataset considering the noisy environment, the TAU Sound Events dataset was mixed with the DNS Challenge Noise-full band dataset, whereas the real noisy data were built by mixing with the real recorded noise dataset. In the mixture data generation process, one of the numerous noise data was selected and mixed in an audio clip of the divided sound event dataset, as described above. The SNR range for the training data consisted of −10, −5, 0, 5, and 10 dB, and that for the validation and test data comprised 2, 0, −2, −5, −7, −10, and −12 dB. The dataset consisted of training, validation, and test sets in a ratio of 8:1:1.

### 3.2. Metrics

To evaluate the SED performance, the segment-based level F-score and error rate were used [45], which were calculated in one-second segments without overlapping. The F-score is calculated using *P* and *R*, where *P* is the precision and *R* is the recall, which are defined as follows:(5)$$P=\frac{\sum_{k=1}^{K}TP\left(k\right)}{\sum_{k=1}^{K}TP\left(k\right)+\sum_{k=1}^{K}FP\left(k\right)},\;R=\frac{\sum_{k=1}^{K}TP\left(k\right)}{\sum_{k=1}^{K}TP\left(k\right)+\sum_{k=1}^{K}FN\left(k\right)}$$
where *K* is the total number of segments, TP(k), FP(k), and FN(k) denote the total number of true positives, false positives, and false negatives in the *k*th one-second segment, respectively. Subsequently, the F-score is calculated as follows:(6)$$F=\frac{2\;\cdot \;P\;\cdot \;R}{P+R}$$

In addition, error rate was measured by calculating the total number of insertions (*I*), deletions (*D*), and substitutions (*S*) relative to the number of active sound events in the reference, *N*, and each is defined as follows:(7)$$\begin{array}{c} S(k)=\min(FN(k),FP(k)), \end{array}$$(8)$$\begin{array}{c} \;D(k)=\max(0,FN(k)-FP(k)), \end{array}$$(9)$$\begin{array}{c} I(k)=\max(0,FP(k)-FN(k)) \end{array}$$

Thus, the error rate is calculated as:(10)$$ER=\frac{\sum_{k=1}^{K}S\left(k\right)+\sum_{k=1}^{K}D\left(k\right)+\sum_{k=1}^{K}I\left(k\right)}{\sum_{k=1}^{K}N\left(k\right)}$$
where N(k) is the number of sound events active in segment *k*.

### 3.3. Feature Extraction for SED Model

It is important to extract feature that can be used efficiently in one domain or in the process of converting to another domain and this technique has also been applied to several different fields [46,47,48,49]. In this study, as the input feature for the SED network, we used a log-mel spectrogram. We applied a window length of 40 ms, hop length of 20 ms, Hanning window size of 40 ms, and fast Fourier transform size of 2048 points. The log-mel spectrogram was extracted as a 128-dimensional feature. We obtain 500 frames in a single 10-s long dataset for the NS, auxiliary, and SED models before the joint training, and by slicing whole the frame in a half, 250 frames in a single 5-s long dataset were used for the cascaded model in the joint training. The input to the SED model was a 128×T log-mel spectrogram feature map, in which *T* denotes the number of frames and is set to 64.

### 3.4. Baseline Model

To verify the SED performance of the proposed system, a baseline model was established by training with noisy data. The baseline model was a CRNN-based classification model, with the same configuration as the auxiliary and SED models. We set the baseline as described above to compare the results based on the joint training and the attention mechanism.

### 3.5. Training Details and Evaluation

This section describes the NS model, auxiliary model in the NS model training procedure, SED model, and joint training.

#### 3.5.1. Noise Suppression Model

The configuration of the NS model, i.e., GC3-TCN, was set using the hyperparameters and notations (described in Table 1) in [29,37] as follows: N=256,L=96,H=128,P=3,X=2,R=4,K=4,M=16,C=32,B=24. The number of channels in the bottleneck and residual paths of the one-dimensional (1-D) conv-blocks and the number of channels in the skip-connection paths of 1-D conv-blocks was 128 each. The NS model was trained for 200 epochs with a deep feature loss and the Adam optimizer at a learning rate of 0.0001. An early stopping method was applied when no best validation model was obtained for 15 consecutive epochs.

#### 3.5.2. Auxiliary Model and Sound Event Detection Model

The two models had the same training procedure and configuration, with differences only in the input data, regardless of them being noisy or clean. The configuration of the classification model is summarized in Table 2. Both models were trained for 150 epochs with binary cross entropy loss and the Adam optimizer at a learning rate of 0.001. A dropout rate of 0.5 was applied to the output of the last convolutional layer and the Bi-LSTM layer. An early stopping method was also applied when no best validation model was found for 10 consecutive epochs.

#### 3.5.3. Joint Training Model

As described in Section 2.3, the joint training model is a fine-tuning process that combines the NS and SED models. In this process, to prevent overfitting and increase the performance of the classifier output, the weights of some layers were frozen, and an attention mechanism was applied to reflect the estimated enhanced information during learning. The hyperparameters of the NS model, SED model, and feature extraction process are the same as during the pretrained model construction. The combined model was trained with binary cross entropy loss and the Adam optimizer at a learning rate of 0.0001. An early stopping method was applied equally.

#### 3.5.4. Evaluation

In the evaluation, we compared the performance of three cases. First, using noisy data as the input, we compared the results of simply combining the NS model at the front end without joint training against those of the baseline model. Subsequently, we compared the before and after joint training results. Finally, based on the range of layer’s weight freezing and the application of the attention mechanism, we compared the performance with those of the previous cases.

## 4. Results and Discussion

### 4.1. Simulation Results

Table 3 summarizes the results of using noisy simulated data mixed with the DNS Challenge Noise-full band dataset and the TAU Spatial Sound Events 2019-Microphone Array Development dataset. The F-score of simply combining the NS and SED models before the joint training was 2.3% lower than the baseline performance. Noise suppression aims to achieve the enhanced signal close to the original clean audio. The model was trained on a different loss function at the pretraining process independently from the SED model. This mismatch between the NS and SED model leads to sub-optimum and degrades the SED performance at the back end with some possible distortions in the audio signal when simply combining both models [30,34]. This results also exhibited the effectiveness of combining both models and optimized on the final SED loss. After joint training with the SED loss, as expected, the F-score and the error rate were improved by 8.7% over the baseline performance and by approximately 0.08, respectively. In addition, the result of the joint training by freezing the weights of the last dense layer resulted in a 1.2% F-score improvement compared to that without freezing. Furthermore, the effectiveness of using the attention module was investigated by comparing the results obtained by the proposed algorithm without/with the attention module. The proposed method with the attention module achieved a superior performance that without the attention module, which verifies its usefulness in the joint training procedure. Thus, the joint training with the NS model at the front end is effective for increasing the SED performance in a low SNR environment.

Figure 4 presents the results of each case per SNR of the test data. Although the total dataset ratio was fixed, the number of datasets for each SNR was not exactly the same because the SNR was randomly selected while creating noisy data. Therefore, there is a possibility that a certain SNR may have been formed high in the evaluation as many datasets were involved in training, and some may have been formed low. However, as a result, when looking at each SNR, it showed improved classification performance by combining the NS at the front end and joint training as expected. In addition, it was shown that the fact that the performance was improved by fixing the weights of the dense layer and applying attention during the joint training process did not change. Furthermore, for the unseen SNR conditions, the proposed method was effective for improving the SED performance.

### 4.2. Real Experimental Results

Table 4 summarizes the results of the noisy data created by mixing the real recorded noise data obtained using a robot cleaner with the TAU Spatial Sound Events 2019- Microphone Array Development dataset. Similar to the simulated results, joint training with NS achieved better performance in terms of the F-score and lower error rate than the baseline method. In addition, as a result of freezing the weights of the dense layer and applying the attention mechanism, the F-score increased by 8.6% and 10%, respectively. This demonstrated that combining NS at the front end to improve the SED performance in a noisy environment leads to enhanced results with both simulation and real data.

## 5. Conclusions

In this paper, we proposed combining a time-domain sound separation-based NS model, GC3-TCN, with a SED model, a CRNN, for noise-robust classification. First, the two models were trained for NS and SED, respectively. Subsequently, a joint model was constructed by combining the two pretrained models. The combined model was jointly fine-tuned with the final SED loss. In the joint training procedure, the dense layer was frozen, and an attention mechanism was added to reflect the enhanced features passed through the NS model in the training. We tested the proposed method on both simulation and real recorded datasets, which showed that using the DNN-based NS at the front end is effective for achieving noise-robust classification.

As for future works, we plan to employ the soft parameter sharing method, which is widely used in multi-task learning in the field of sound event detection and localization, during the joint training process instead of the attention mechanism. Additionally, we consider using the knowledge distillation, also known as a teacher-student method, to further improve the noise-robust classification.

## Figures and Tables

**Figure 1 sensors-21-06718-f001:**
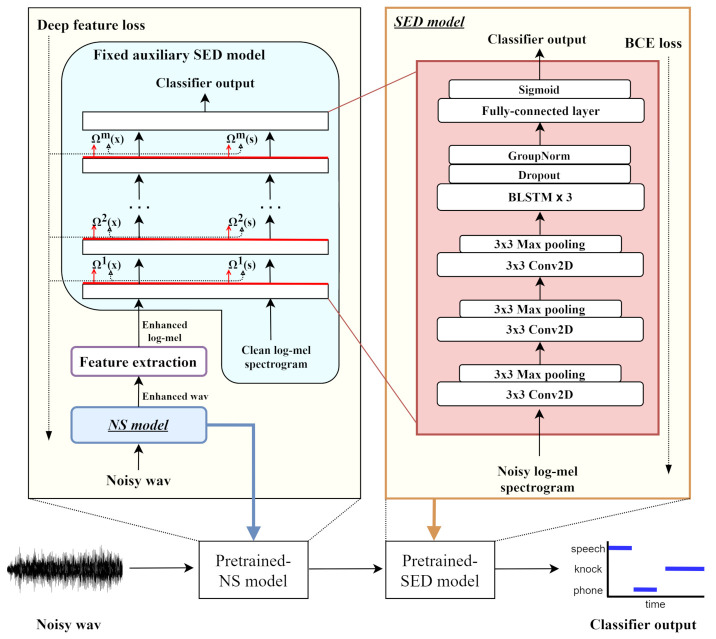
Overall structure showing pretraining process before joint training, where *s* and *x* denote the clean and enhanced features, respectively. Additionally, *m* is the number of layers, $\Omega^{m}$(·) is the activation feature output of the $m^{th}$ layer, and classifier output is the event presence probability output.

**Figure 2 sensors-21-06718-f002:**
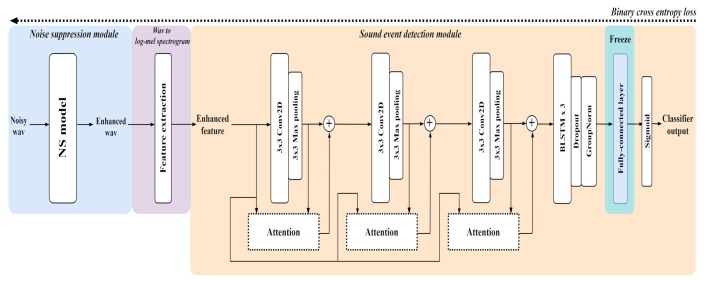
Overall procedure of attention-based joint training.

**Figure 3 sensors-21-06718-f003:**
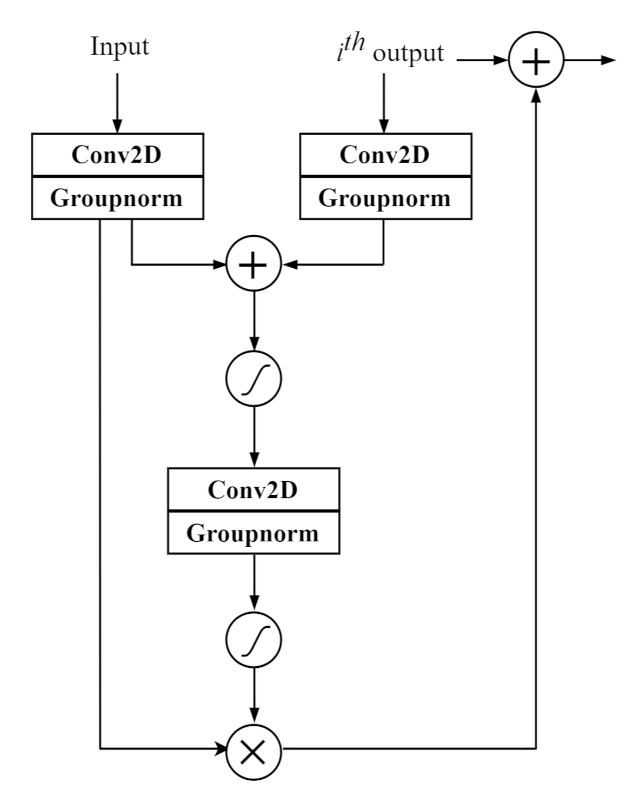
Attention mechanism.

**Figure 4 sensors-21-06718-f004:**
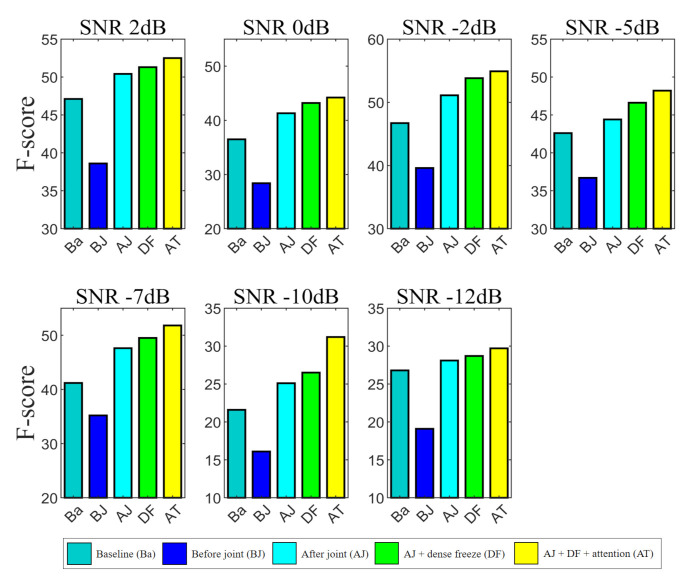
Performance comparison of the test data in different cases based on joint training method and use of additional attention mechanism under different SNR environments. SNR levels of test data are 2, 0, –2, –5, –7, –10, and –12 dB.

**Table 1 sensors-21-06718-t001:** Hyperparameters and notations used in GC3-TCN.

Hyperparameters	Notation
Number of encoder filters	N
Length of the filters	L
Number of channels in convolutional blocks	H
Kernel size in convolutional blocks	P
Number of convolutional blocks in each repeat	X
Number of repeats	R
Number of groups	K
Group size	M
Context size (in frames)	C
TCN block size (in frames)	B

**Table 2 sensors-21-06718-t002:** Configuration of CRNN.

Layers	Output Size
Input	1 × 128 ×T
Convolution 1	16 × 64 × T
Max-pooling 1	16 × 32 × T
Convolution 2	32 × 16 × T
Max-pooling 2	32 × 8 × T
Convolution 3	64 × 4 × T
Max-pooling 3	64 × 2 × T
Reshape	T × 128
BLSTM × 3	T × 128
Fully connected	T × 128
Output	T × 11

**Table 3 sensors-21-06718-t003:** Performance comparison using the simulated noise dataset for input mixture. JT denotes joint training.

Method	Model		Evaluation	
	NS	SED	F-Score (%)	Error Rate
Baseline	-	CRNN	33.7	0.78
Before JT (pretrained)	GC3-TCN	CRNN	31.4	0.81
After JT (fine-tuned)	GC3-TCN	CRNN w/o attention		
		w/o freeze	42.4	0.70
		freeze dense	**43.6**	**0.68**
		+ CNN	38.0	0.74
		+ RNN	34.3	0.78
		CRNN w/ attention		
		+ freeze dense	**45.2**	**0.66**

**Table 4 sensors-21-06718-t004:** Performance comparison using the real recorded noise dataset for input mixture. JT denotes joint training.

Method	Model		Evaluation	
	NS	SED	F-Score (%)	Error Rate
Baseline	-	CRNN	41.9	0.71
Before JT (pretrained)	GC3-TCN	CRNN	36.0	0.75
After JT (fine-tuned)	GC3-TCN	CRNN w/o attention		
		w/o freeze	48.7	0.66
		freeze dense	**50.5**	**0.64**
		+ CNN	47.8	0.66
		+ RNN	44.6	0.69
		CRNN w/ attention		
		+ freeze dense	**51.9**	**0.62**

## Data Availability

Not applicable.

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
