# Peer review of "Attention-Based Joint Training of Noise Suppression and Sound Event Detection for Noise-Robust Classification"

_sensors, 2021, doi:10.3390/s21206718_

Round 1

Reviewer 1 Report

This paper describes a sound event detection system consisting of a noise suppression model and an event classifier trained together. Joint training of different models is not new, but there are some original parts in the implementation. Overall, the paper is well written and technically sound, tough some parts need improvement and more detailed description. The main drawback is the experimental setup. Authors compare several versions of the same system, but not with any other existing method. They should either do that or evaluate their system on some benchmark dataset, DCASE series of challenges, for example, to position themselves among other players.

Some comments and corrections:

1. Section 2, the first sentence: "...system, which consists of pretraining,  joint training,..." A system consists of modules, blocks, not training procedures.

2. Fig.1: activation output is denoted by "big omega", but in Eq.4 as "alpha".

3. Fig.1, Fig.2: max pooling should be 1x2, not 3x3.

4. Section 2.4: attention mechanism is poorly described. What is the Groupnorm purpose? How are the different input and output sizes handled?

5. Fig.1 is missing the Dropout layer.

6. Section 3.1: audio events in the data are insufficiently described. What is their duration? How many events are there in one segment? Do they overlap?

7. Section 3.3: The input window is 64 frames. Then how are 5/10 sec segments processed - with a sliding window (if yes, what is the shift?) or not?

8. Section 3.2: Instead of counting insertions, deletions, and substitutions, for SAD tasks segment-based and event-based evaluation are common (look at "Sound Event Detection" paper from IEEE Signal Proc. Mag., Sept. 2021)

Author Response

We appreciate very much the valuable comments and suggestions of the reviewers on our paper. In our revised manuscript, we have incorporated all the comments and suggestions made by the reviewers, and have given additional explanations. Our detailed responses are as follows. Please see the attachment.

Reviewer 2 Report

The reviewer's comments can be found in the updated pdf file.

Author Response

(The authors gave the same response as above.)

Reviewer 3 Report

The paper is quite interesting and is relevant to the noise suppression and sound event detection community. For a higher interest to noise suppression audience, the following papers should be cited and few lines are added in the Introduction section and Noise suppression models in section 2.1 where problems of noise suppression should be highlighted and how others authors have employed Non-Gaussian solutions. Following papers and references in them might help to highlight the problem associated with Noise suppression

1) Q. Z. Ahmed, M. Alouini and S. Aissa, "Bit Error-Rate Minimizing Detector for Amplify-and-Forward Relaying Systems Using Generalized Gaussian Kernel," in IEEE Signal Processing Letters, vol. 20, no. 1, pp. 55-58, Jan. 2013, doi: 10.1109/LSP.2012.2229271.

2) M. Novey, T. Adali and A. Roy, "A Complex Generalized Gaussian Distribution— Characterization, Generation, and Estimation," in IEEE Transactions on Signal Processing, vol. 58, no. 3, pp. 1427-1433, March 2010, doi: 10.1109/TSP.2009.2036049.

3) Yang Liu, Tianshuang Qiu, Jingchun Li, "Joint estimation of time difference of arrival and frequency difference of arrival for cyclostationary signals under impulsive noise", Digital Signal Processing, vol. 46, pp. 68, 2015.

Furthermore, in section 3.3 feature extraction is mentioned. It is worth mentioning that this technique has also been applied to a number of different fields such as communications, signal processing, WSN etc. The following papers and references therein might help to highlight the points

1) J. S. Goldstein and I. S. Reed, "Subspace selection for partially adaptive sensor array processing", IEEE Trans. Aerosp. Electron. Syst., vol. 33, no. 2, pp. 539-544, Apr. 1997.

2) M.-S. Alouini, A. Scaglione and G. B. Giannakis, "PCC: Principal components combining for dense correlated multipath fading environments", Proc. IEEE 52nd VTC-Fall, pp. 2510-2517, 2000-Fall.

3)R. Husbands, Q. Ahmed and J. Wang, "Transmit antenna selection for massive MIMO: A knapsack problem formulation," 2017 IEEE International Conference on Communications (ICC), 2017, pp. 1-6, doi: 10.1109/ICC.2017.7996694.

4) R. C. de Lamare and R. Sampaio-Neto, "Adaptive reduced-rank equalization algorithms based on alternating optimization design techniques for MIMO systems", IEEE Trans. Veh. Technol., vol. 60, no. 6, pp. 2482-2494, Jul. 2011.

Author Response

(The authors gave the same response as above.)

Round 2

Reviewer 1 Report

The revised version addresses the main concerns I had before and I think the paper can be published in its present form.